# Influence of Heat Treatments on Microstructure and Mechanical Properties of Ti–26Nb Alloy Elaborated In Situ by Laser Additive Manufacturing with Ti and Nb Mixed Powder

**DOI:** 10.3390/ma12010061

**Published:** 2018-12-25

**Authors:** Jing Wei, Hongji Sun, Dechuang Zhang, Lunjun Gong, Jianguo Lin, Cuie Wen

**Affiliations:** 1School of Materials Science and Engineering, Xiangtan University, Xiangtan 411105, China; weijing19930624@163.com (J.W.); sun_hong_ji@163.com (H.S.); 2Key Laboratory of Materials Design and Preparation Technology of Hunan Province, Xiangtan University, Xiangtan 411105, China; ljgong@xtu.edu.cn; 3School of Aerospace, Mechanical and Manufacturing Engineering, RMIT University, Melbourne 3083, Australia; cuie.wen@rmit.edu.au

**Keywords:** heat treatment, in situ alloying, laser additive manufacturing, mechanical properties, microstructure, Ti–Nb alloy

## Abstract

In the present work, a Ti–26Nb alloy was elaborated in situ by laser additive manufacturing (LAM) with Ti and Nb mixed powders. The alloys were annealed at temperatures ranging from 650 °C to 925 °C, and the effects of the annealing temperature on the microstructure and mechanical properties were investigated. It has been found that the microstructure of the as-deposited alloy obtained in the present conditions is characterized by columnar prior β grains with a relatively strong <001> fiber texture in the build direction. The as-deposited alloy exhibits extremely high strength, and its ultimate tensile strength and yield strength are about 799 MPa and 768 MPa, respectively. The annealing temperature has significant effects on the microstructure and mechanical properties of the alloys. Annealing treatment can promote the dissolution of unmelted Nb particles and eliminate the micro-segregation of Nb at the elliptical-shaped grain boundaries, while increasing the grain size of the alloy. With an increase in annealing temperature, the strength of the alloy decreases but the ductility increases. The alloy annealed at 850 °C exhibits a balance of strength and ductility.

## 1. Introduction

Beta (β) titanium (Ti) alloys containing non-cytotoxic elements such as niobium (Nb), zirconium (Zr), tantalum (Ta), tin (Sn), and molybdenum (Mo) have been intensively studied for applications in biomedical domains [1,2,3,4,5]. Among these, Ti–Nb alloys have attracted particular attention due to their high strength (600 MPa) and their very low elastic modulus (50 GPa), close to that of cortical bone (30 GPa), which was observed to reduce the stress-shielding phenomenon [6]. These alloys are considered to be potential substitute materials for conventional materials such as Ti–Ni or Ti–6Al–4V in order to prevent the release of toxic nickel (Ni), aluminium (Al), or vanadium (V) ions into the human body [7,8,9,10,11].

Recently, laser additive manufacturing (LAM), based on coaxed powder melting and rapid solidification through layer-upon-layer deposition, has attracted a great deal of attention in the fabrication of fully dense near net-shaped metallic components [12,13,14,15]. It has several significant advantages over traditional manufacturing methods, such as a wide range of material forms, a simple manufacturing process for complex parts, short production cycle, raw material saving, excellent performance, etc. So it is an effective method for the personal customization of biological implants.

Generally, the powders for LAM are obtained from a prealloyed material that is converted into powder by plasma arc or gas atomization to achieve spherical particles with a particle size between 10 and 110 μm. In the case of Ti–Nb alloys, their high melting temperatures make it very complicated to produce powders in this way. So the Ti–Nb alloy powders that can be applied to direct laser metal-forming are still rare or even commercially unavailable. Recently, some researchers attempted to fabricate Ti–Nb alloys using selective laser melting (SLM) with mixtures of elemental powders of pure Ti and Nb. For example, Fischer et al. [16] fabricated Ti–26Nb (at.%) using SLM with elemental Ti and Nb mixed powders with high energy levels, and the alloy exhibited a homogeneous and dense microstructure with a β structure. Wang et al. [17] also fabricated Ti–Nb alloys with different Nb content using the same method and investigated the effects of Nb content on the microstructure and mechanical properties; they found that SLM could be used for in situ fabrication of Ti–Nb bone implants with tailored mechanical and biomedical properties by adjusting the level of Nb.

It has been reported that, compared to the microstructure achieved with conventional casting and forging, Ti alloys prepared using LAM usually exhibit a quite different microstructure due to repeated rapid solidification and rapid annealing during laser forming. Moreover, defects can be found in the as-deposited part, such as inconsistency in the structure and stability of mechanical properties, residual stress, and pores, which can weaken the mechanical properties of the alloys. Post-heat treatment can have an important influence on the microstructure and properties of the alloys [18,19,20]. However, microstructure transformation after heat treatment of as-deposited Ti–Nb alloys has rarely been investigated. Therefore, a deeper understanding of the microstructural evolution of Ti–Nb formed by LAM during heat treatment would allow improvement of the mechanical mechanical of these alloys.

In the present work, a Ti–26at.%Nb alloy was prepared by directly melting a mixture of elemental Ti and Nb powders under a laser beam. The alloy was annealed at temperatures ranging from 650 °C to 925 °C and the influence of the annealing temperature on the microstructure evolution and mechanical properties of the alloy was investigated.

## 2. Experimental Methods

### 2.1. Material Manufacturing

Commercial gas-atomized pure Ti powder (purity 99.99%, Changsha Tianjiu Metal Materials Co. Ltd., Changsha, China) with a particle size of 45–105 μm and pure Nb powder (purity 99.99%, Changsha Tianjiu Metal Materials Co. Ltd., Changsha, China) with a particle size of 48–75 μm were used as the raw materials. The two powders were mixed in the weight ratio Ti:Nb = 59.5:40.5 (namely 74:26 in atomic ratio) by ball milling for 1 h. Figure 1 shows the morphologies of the Ti and Nb powders and the mixtures. It is clear that the Ti powder has regular spheres with the presence of satellites (Figure 1a), whereas the Nb has irregular shapes (Figure 1b). After mixing, the Nb particles were uniformly distributed around the Ti particles (see Figure 1c). The melting point of Nb (2477 °C) is much higher than that of Ti (1668 °C). To facilitate better melting of the Nb and its faster diffusion in Ti, Nb powder with a smaller particle size than the Ti powder was used in the present work.

A square block sample of the obtained Ti–26Nb alloy with dimensions of 400 mm × 350 mm × 230 mm (see Figure 2) was fabricated by a LAM process on a YLS-4000-CL machine (IPG photonics corporation, Oxford, MA, USA), in which a fiber laser was produced by the IPG photonics corporation and the powder was fed in coaxial feeding mode with argon as the carrier gas. SD, LD, and BD represent the scanning direction, lateral direction, and build direction, respectively. The laser-deposition processing parameters were: laser nominal output power 750 W; laser beam diameter 2.5 mm; scanning speed 480 mm/min; and powder feed rate 2.2 g/min.

### 2.2. Heat Treatment

The samples for heat treatment were sectioned along the BD with dimensions of 25 mm in length (BD), 10 mm in width (SD), and 1.2 mm in thickness (LD). The samples were vacuum encapsulated in quartz tubes and annealed at temperatures ranging from 650 °C to 925 °C for a duration of 0.5 h followed by water quenching, as listed in Table 1.

### 2.3. Microstructure Characterization

The microstructures of the alloy samples before and after heat treatment were characterized by optical microscopy (OM; BX51M, Olympus, Tokyo, Japan), scanning electron microscopy (SEM; MIRA3 LMU, Tescan, Brno, Czech), and X-ray diffraction (XRD; D/max 2500, Rigaku, Tokyo, Japan). Grain size and grain orientation were determined by electron backscatter diffraction (EBSD; HKL, Oxford, UK). Several regions in the samples were chosen for the EBSD analysis, and average grain size and grain orientations were determined by statistical analysis.

### 2.4. Mechanical Property Testing

Tensile tests of the Ti–26Nb alloy were carried out on an Instron 5569 universal testing machine (Instron, Boston, MA, USA). The samples for the tests were cut from the middle section of the block prepared by LAM with a gauge section of 1 mm × 2.5 mm × 8 mm, the geometry of which is schematically shown in Figure 3. All specimens are shown along the BD. Three samples were measured for each condition in order to reduce measurement error.

For comparison, another Ti–26Nb alloy was prepared using a conventional arc melting method with pure Ti and Nb as the raw materials. The ingot was hot-rolled by 90% in thickness, then tensile testing of the mechanical properties according to the above method was also carried out in the same conditions.

## 3. Results and Discussion

### 3.1. Phase Composition

Figure 4 shows the X-ray diffraction (XRD) patterns of the Ti–26Nb alloy fabricated by LAM and after annealing at different temperatures for 0.5 h. It can be seen that the as-deposited alloy shows a single β phase with a body-centered cubic (bcc) structure. After annealing at 650 °C for 0.5 h, diffraction peaks from an α phase could be seen in the XRD pattern of the alloy, implying that an α phase precipitated in the alloy. As the annealing temperature increased to 925 °C, the peaks from the α phase completely disappeared. By careful comparison, it can be seen that the diffraction peaks of the β phase on the XRD pattern of the alloy after annealing have slightly shifted to the left relative to the as-deposited sample (as shown in Figure 4). This implies that more Nb atoms have dissolved in the Ti lattice. This is because the radius of the solute atom Nb is slightly larger than that of the Ti atom, and thus a higher concentration of Nb leads to an increase in the lattice constant of the β phase.

### 3.2. Microstructure

Figure 5 illustrates the microstructure of the as-deposited Ti–26Nb alloy. It is clear that multilayer deposits with uniform thickness and regular distribution have been formed in the as-deposited Ti–26Nb alloy. The prior β phase with columnar grains oriented more or less in the BD, which penetrate the multilayer cladding layer, can be observed (see Figure 5a). The special formation mechanism of columnar grains has been clearly explained in previous work [21]. Some unmelted Nb can be observed in the Ti–Nb alloy due to its relatively high melting temperature of 2477 °C, as shown in Figure 5a. The results indicate that the applied energy density of the LAM process is sufficient to completely melt the Ti powder, but some of the larger Nb particles are only partially melted. Similar behavior was also found in a previous report [22]. Moreover, some pores can be seen on the surface of the sample, which may have been caused by ball formation and gas inclusion generated during the melting and remelting process [23].

A magnified image reveals that some ultra-fine elliptical-shaped grains (dendritic grains) with dimensions of ~10 μm in width and ~80 μm in length were formed in the representative regions (see Figure 5b). This may be attributed to the fact that the solidified layer acts as a substrate for the solidification of the melt, leading to the formation of elliptical-shaped grains perpendicular to the solidification front [24]. Moreover, the elliptical grains grew in a wavelike fashion and their boundaries are poor in Nb, as seen in Figure 5c,d.

Figure 6 shows the distributions of Ti and Nb elements in the molten pool of Ti-26Nb alloy. It can be seen that the Ti and Nb content at the melt pool boundaries (MPBs) is identical to that inside of the molten pool, implying that the solidification of the molten pool is completed in a short time, which is very different from conventional casting. As a result, the Nb particles do not have enough time to sink to the bottom of the molten pool and thus remain.

To investigate the effects of the heat treatment on the microstructure of the alloy fabricated by LAM, the as-deposited alloy samples were annealed at temperatures ranging from 650 °C to 925 °C. Figure 7 illustrates the microstructures of the samples after annealing at the different temperatures.

For the sample annealed at 650 °C for 0.5 h, elliptical grains began to grow and their size distribution became more uniform (as shown in Figure 7a) as compared to as-deposited sample. Close observation reveals that the some fine acicular secondary α_S_ phases precipitated in the areas poor in Nb atoms in the alloys annealed at 650 °C and 750 °C, as shown in the inset of Figure 7c,d). As the annealed temperature increased to 850 °C, the boundaries of the ellipses became unclear, implying the dissolution of the dendritic grains due to the diffusion of Nb atoms. Moreover, almost all the α_S_ phases disappeared in the alloys as the annealing temperature increased over 850 °C, and the alloy that annealed at 925 °C exhibited an even microstructure with all β phases (see Figure 7e,f). This is consistented with the XRD analysis.

Figure 8 illustrates the Nb concentration in the elliptical-shaped grains and at the grain boundaries as obtained by energy-dispersive X-ray spectroscopy (EDS) analysis. It can be seen that, for the as-deposited alloy, the Nb content at the grain boundaries area is much lower than in the elliptical grains. After annealing at 650 °C, the Nb content at the grain boundaries increased to become close to that inside the grains, implying the diffusion of Nb from inside to the grain boundaries during the annealing treatment. As the annealing temperature further increased, the Nb content both inside and at the boundaries of the grains slightly increased, implying the dissolution of the unmelted Nb particles.

### 3.3. Grain Orientation and Grain Size

As mentioned above, the microstructure of the as-deposited Ti–26Nb alloy exhibited columnar β grains with fine dendrites inside oriented more or less in the BD. The preferred direction of the dendrite growth in the Ti–Nb alloy can be ascribed to the largest thermal gradient parallel to BD during the LAM process, leading to the formation of this texture [25]. The texture was further verified by EBSD orientation maps, as seen in Figure 9a. It is clear that the coarse columnar β grains of the as-deposited alloy exhibits a relatively strong <100> fiber texture. The annealing treatment tended to make the grain orientation uniform, as shown in Figure 9b,c.

Further confirmation of the preferred orientation is illustrated by the corresponding EBSD pole figure in Figure 10. It can be seen that the as-deposited alloy exhibits the strongest <001> texture. After annealing treatment at 650 °C, the <001> texture of the alloy was weakened. However, with the annealing temperature further increasing to 850 °C, the texture became stronger.

The effects of annealing temperature on the columnar grain size of the alloy were also investigated. Figure 11 shows the grain size distribution of the alloy before and after annealing at different temperatures. It can be seen that the average grain size of the as-deposited alloy is about 143.3 μm.

While annealing treatment leads to clear growth in the grain size of the columnar β grains, with an increase in annealing temperature the size of the columnar β grains increases. The alloy after annealing at 925 °C exhibited the largest grain size, which reached 205.5 μm. The growth in grain size caused by the heat treatment may have decreased the strength of the alloy.

### 3.4. Mechanical Properties

The stress-strain curves of Ti–26Nb alloys after annealing at different temperatures (see Figure 12). The mechanical properties of the alloy before and after annealing at different temperatures were evaluated by Figure 12 and the results are shown in Figure 13 and Table 2. It can be seen that the as-deposited Ti–26Nb sample shows relatively high strength, with the ultimate tensile strength (UTS) and yield strength (YS) about 799 MPa and 768 MPa, respectively, which are much higher than those of the as-rolled Ti–26Nb alloy (see Table 2). The fine-grained structures present inside the β grains produced by the rapid solidification process may be responsible for the high strength [26].

The heat treatment decreased the strength but increased the ductility of the alloy, depending on the annealing temperature. It can be seen that, after annealing at 650 °C for 0.5 h, the UTS and YS of the alloy reduced by about 176 and 223 MPa, respectively, in comparison with the as-deposited alloy due to the growth of the prior columnar β grains and the dissolution of the fine dendrites [26]. It should be noted that, as the annealing temperature further increased, the strength of the alloys gradually increased despite the growth of the grains. The UTS and YS of the alloy annealed at 925 °C were about 722 MPa and 685 MPa, respectively. The increase in strength of the alloy annealed at a relatively high temperature may be attributable to the solid solution strengthening the effect of the Nb solute atoms. As mentioned above, annealing treatment can lead to the dissolution of unmelted Nb atoms into the Ti lattice and, with an increase in annealing temperature, the content of Nb in the β increases. This may cause more severe lattice distortion, which can increase the resistance of the dislocation motion and consequently promote the strength of the alloy.

With regard to the ductility of the alloy, the as-deposited alloy exhibited relatively low ductility with elongation of 14.3%. This may be attributable to the presence of a large amount of residual stress in the as-built condition and defects such as pores, micro-segregation, and unmelted Nb particles in the as-deposited alloy. After annealing at 650 °C, the ductility of the alloy slightly increased due to the elimination of residual stress [27] and microstructural homogenization. However, the precipitation of secondary α phases at the grain boundaries may have deteriorated the ductility of the alloy. As the annealing temperature increased to 850 °C, most unmelted Nb particles dissolved into the β phase, which may have increased its stability, and thus no precipitation of secondary α phases occurred. This led to an increase in the ductility of the alloy. As a result, the alloy annealed at 850 °C exhibited relatively high ductility with elongation of 21.6%. However, with the annealing temperature further increasing to 925 °C, the grains of the β phase became coarse, which may have decreased the ductility of the alloy, as its elongation decreased to 18.9%.

## 4. Conclusions

In this work, a Ti–26Nb alloy was successfully prepared from a mixture of titanium and niobium powders by laser additive manufacturing. The as-deposited alloy was annealed at different temperatures ranging from 650 °C to 925 °C, then the microstructure evolution and tensile mechanical properties were evaluated. The main conclusions can be summarized as follows:

(1) The microstructure of the as-deposited Ti–26Nb alloy was characterized by prior columnar β grains with a relatively strong <001> fiber texture due to a large temperature gradient and remelting penetration in the build direction. Defects such as pores, unmelted Nb particles, and micro-segregation at the grain boundaries of the elliptical-shaped grains could be observed in the as-deposited Ti–26Nb alloy. Its ultimate tensile strength and yield strength were about 799 MPa and 768 MPa, respectively, much higher than those of hot-rolled Ti–26Nb (428 MPa and 415 MPa, respectively).

(2) The heat treatment had an important influence on the microstructures of the as-deposited alloy, depending on the annealing temperature. After annealing at 650 °C for 0.5 h, many fine acicular secondary α_S_ phases precipitated at the boundaries of the elliptical-shaped grains with poor Nb content. With the annealing temperature increasing, the unmelted Nb particles dissolved into the Ti lattice, leading to an increase in Nb concentration in the β matrix, and the composition of the alloy tended to become uniform by the diffusion of Nb atoms. Furthermore, the annealing treatment led to growth in the size of the grains.

(3) Heat treatment decreased the strength but increased the ductility of the alloy. After annealing at 650 °C for 0.5 h, the prior columnar β grain growth and fine dendrite dissolution resulted in the ultimate tensile strength and yield strength decreasing to 623 MPa and 543 MPa, respectively. The increase in annealing temperature slightly increased the strength of the alloy. The solution strengthening due to the dissolution of the unmelted Nb particles together with the coarsening of the β grains may be responsible for the changing trend of the mechanical properties of the alloy with annealing temperature. The alloy annealed at 850 °C for 0.5 h exhibited a good balance of strength and ductility.

## Figures and Tables

**Figure 1 materials-12-00061-f001:**
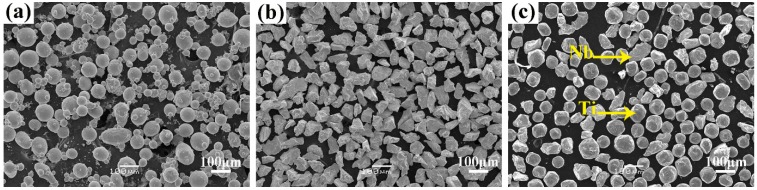
Scanning electron microscope (SEM) images of feedstock powders: (**a**) pure Ti; (**b**) pure Nb; and (**c**) Ti–26Nb powder mixture.

**Figure 2 materials-12-00061-f002:**
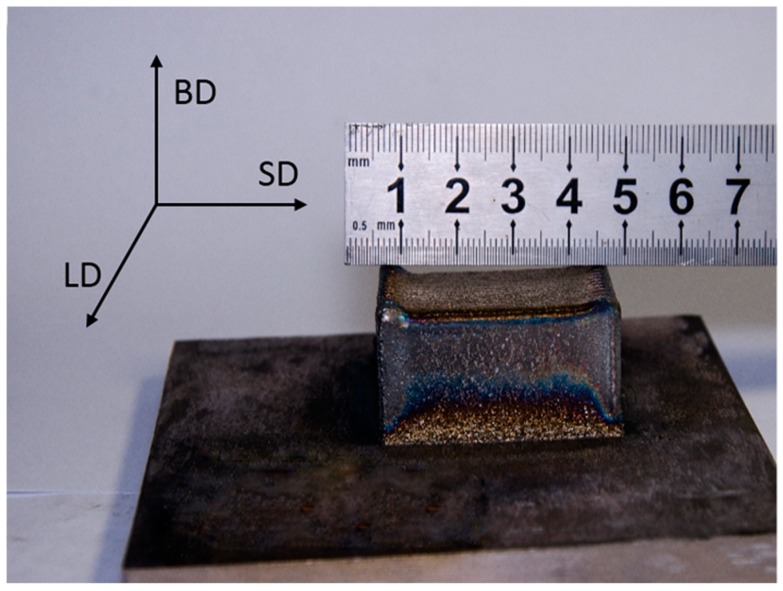
Bulk Ti–26Nb sample fabricated by laser additive manufacturing (LAM).

**Figure 3 materials-12-00061-f003:**
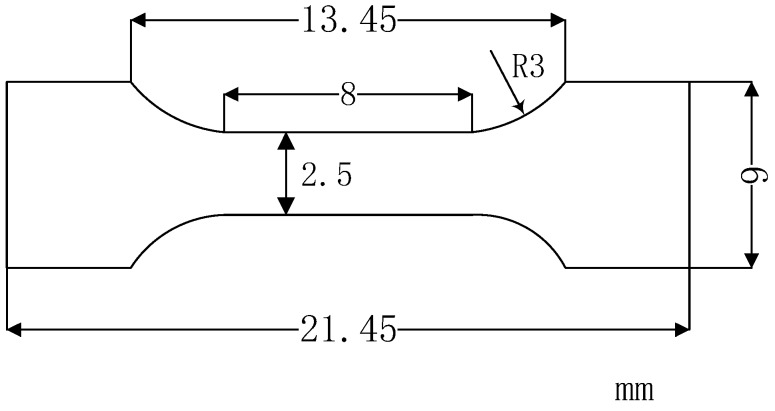
Schematic geometry of tensile specimen.

**Figure 4 materials-12-00061-f004:**
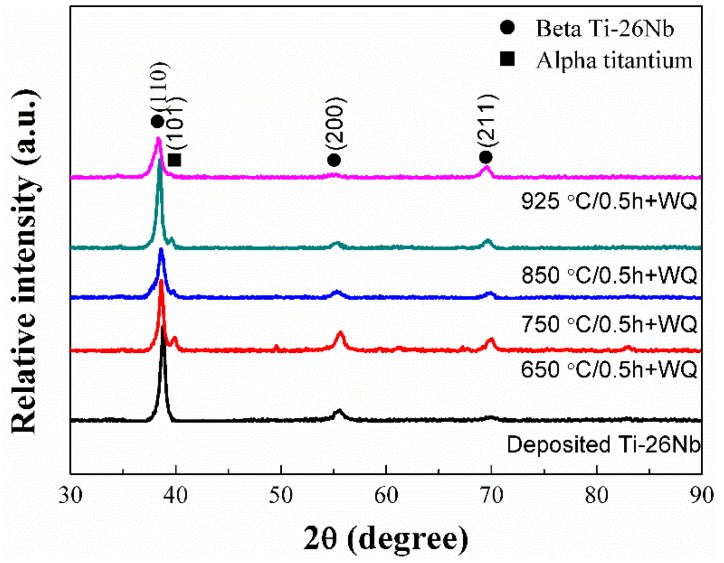
X-ray diffraction (XRD) patterns of LAM-processed Ti–26Nb alloys after annealing treatment at different temperatures.

**Figure 5 materials-12-00061-f005:**
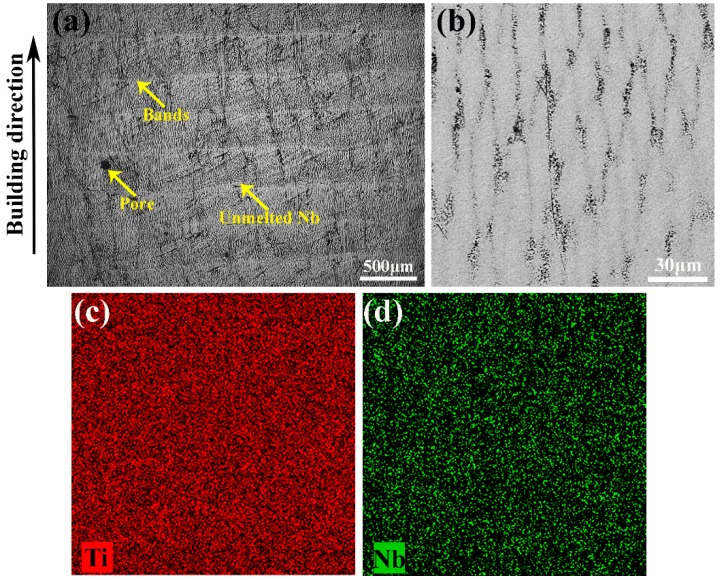
(**a**) and (**b**) Microstructures of as-deposited Ti–26Nb alloy; (**c**) and (**d**) elemental mapping images of as-deposited Ti–26Nb of (**b**).

**Figure 6 materials-12-00061-f006:**
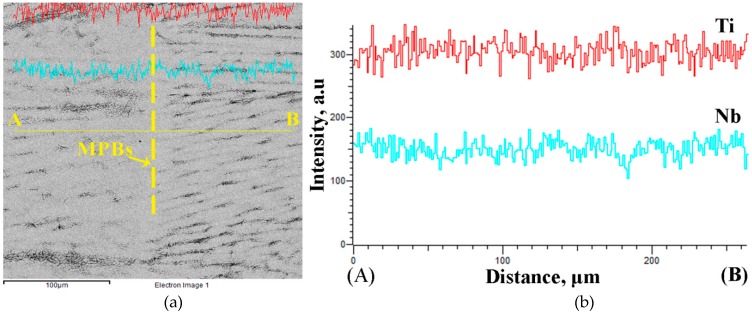
(**a**) Energy-dispersive X-ray spectroscopy (EDS) line scanning of melt pool in lateral plane of Ti–26Nb alloy (**b**) corresponding line scanning results of Ti and Nb elements.

**Figure 7 materials-12-00061-f007:**
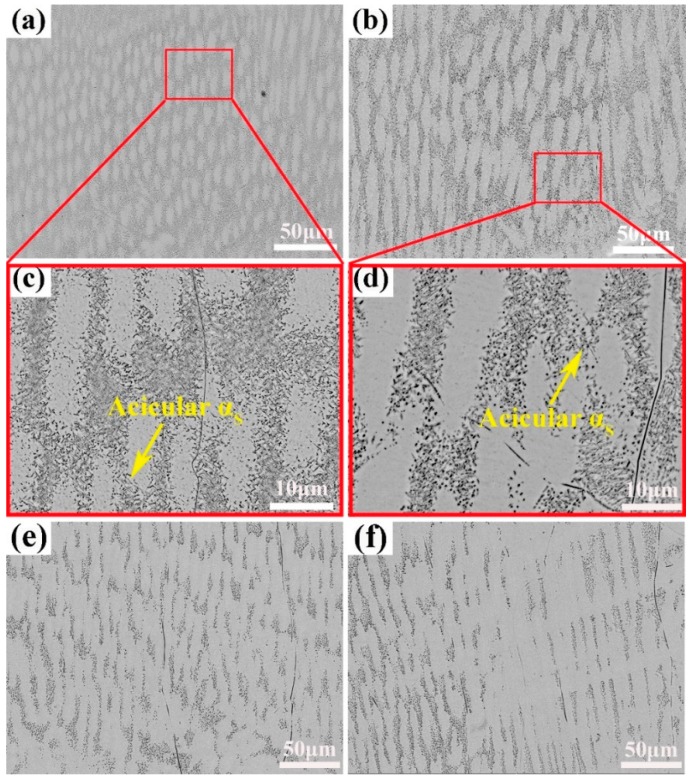
Microstructures of annealed samples at different temperatures for 0.5 h: (**a**) 650 °C; (**b**) 750 °C; (**c**) and (**d**) corresponding high-magnification morphologies of (**a**) and (**b**), respectively; (**e**) 850 °C; (**f**) 925 °C.

**Figure 8 materials-12-00061-f008:**
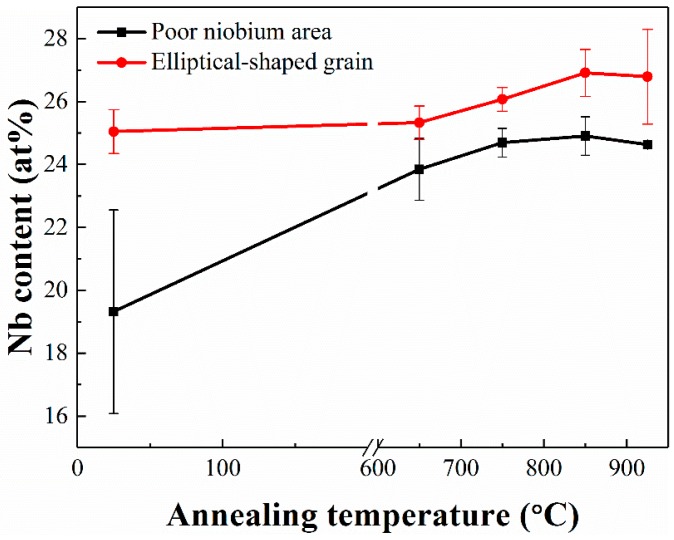
Comparison of Nb content in different regions of Ti–26Nb alloys after annealing at different temperatures.

**Figure 9 materials-12-00061-f009:**
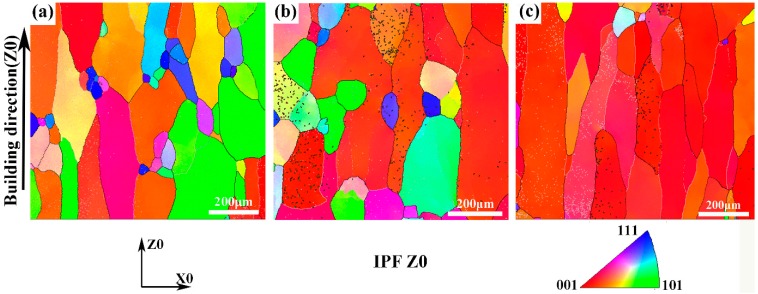
IPFZ electron backscatter diffraction (EBSD) map presented with respect to z-direction showing multiple grain orientations: (**a**) as-deposited Ti–26Nb alloy; (**b**) after annealing at 650 °C; (**c**) after annealing at 850 °C.

**Figure 10 materials-12-00061-f010:**
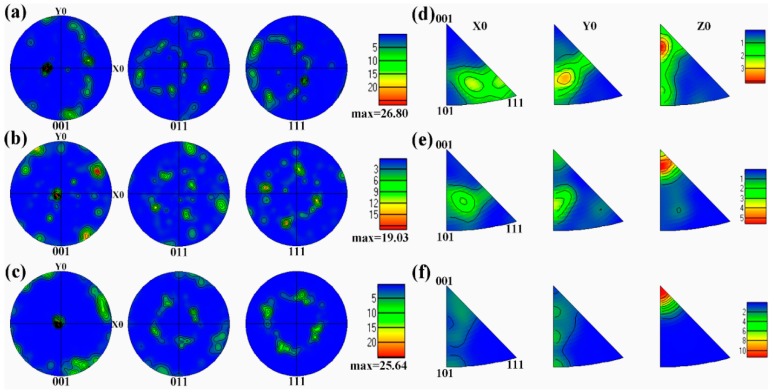
EBSD pole figure and inverse pole figure of Ti-26Nb samples: (**a**) and (**d**) as-deposited Ti–26Nb alloy; (**b**) and (**e**) after annealing at 650 °C; (**c**) and (**f**) after annealing at 850 °C.

**Figure 11 materials-12-00061-f011:**
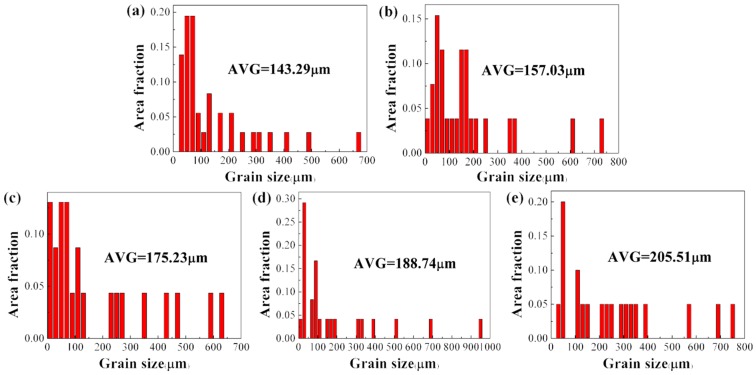
Grain size distributions of specimens: (**a**) as-deposited Ti–26Nb alloy; (**b**) after annealing at 650 °C; (**c**) after annealing at 750 °C; (**d**) after annealing at 850 °C; (**e**) after annealing at 925 °C.

**Figure 12 materials-12-00061-f012:**
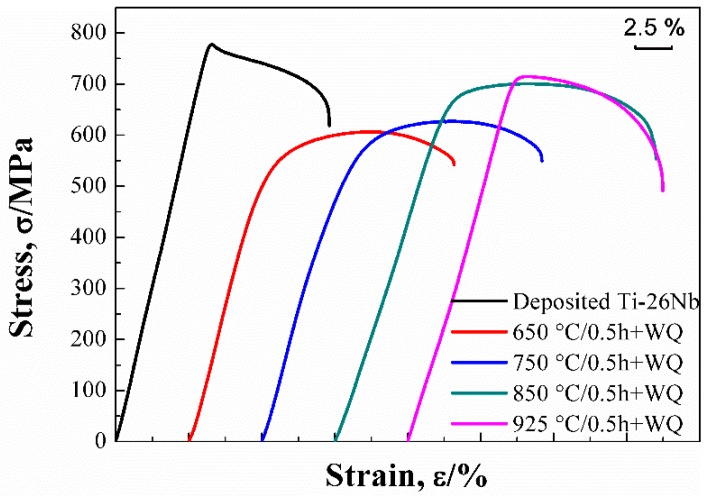
Stress-strain curves of Ti–26Nb alloys after annealing at different temperatures.

**Figure 13 materials-12-00061-f013:**
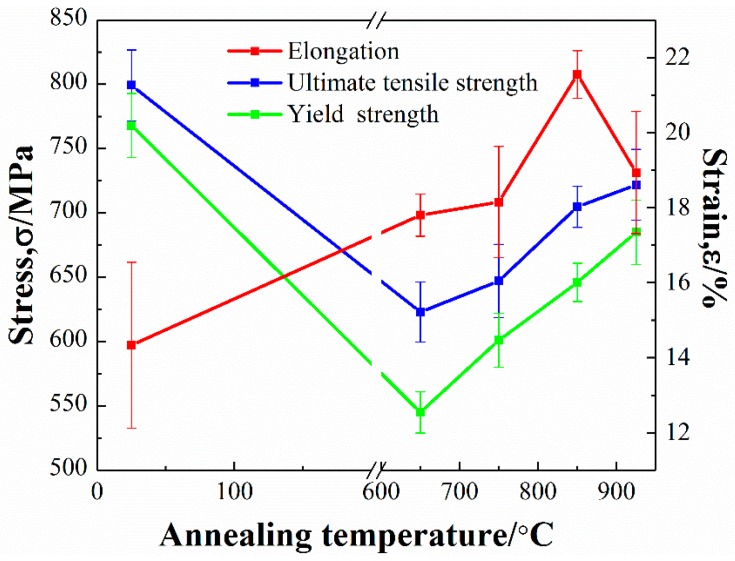
Mechanical properties of Ti–26Nb alloys after annealing at different temperatures.

**Table 1 materials-12-00061-t001:** Details of heat treatment for LAM Ti–26Nb alloys.

Sample No.	Heat Treatment
1	650 °C/0.5 h + WQ
2	750 °C/0.5 h + WQ
3	850 °C/0.5 h + WQ
4	925 °C/0.5 h + WQ

Note: WQ is water quenching.

**Table 2 materials-12-00061-t002:** Mechanical properties of Ti–Nb alloys fabricated by LAM and casting.

State of Materials	Ultimate Tensile Strength/MPa	Yield Strength/MPa	Elongation/%
As–deposited	799 ± 27	768 ± 25	14.3 ± 2.2
650 °C/0.5 h + WQ	623 ± 23	545 ± 16	17.8 ± 0.6
750 °C/0.5 h + WQ	647 ± 28	601 ± 21	18.2 ± 1.5
850 °C/0.5 h + WQ	704 ± 16	646 ± 15	21.6 ± 0.6
925 °C/0.5 h + WQ	722 ± 27	685 ± 25	18.9 ± 1.6
Hot–rolled of cast Ti–26Nb	428 ± 15	415 ± 12	18.2 ± 0.8

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
