# Peer review of "Influence of Heat Treatments on Microstructure and Mechanical Properties of Ti–26Nb Alloy Elaborated In Situ by Laser Additive Manufacturing with Ti and Nb Mixed Powder"

_materials, 2018, doi:10.3390/ma12010061_

Round 1

Reviewer 1 Report

Author needs to address the following aspects:

1. Particle size distribution. There are challenges when there is a wide range of powder particle size distribution (45–105 μm) – especially when mixed with another powder. Author needs to provide justification that this powder results in uniform material distribution.

2. Process parameters are given. Knowing that this is a new powder composition, evidence that these processing parameters are reasonable, a special study needs to be presented.

3. Stress-strain curves should be included.

Author Response

Point 1. Particle size distribution. There are challenges when there is a wide range of powder particle size distribution (45–105 μm) – especially when mixed with another powder. Author needs to provide justification that this powder results in uniform material distribution.

Response: Thank you very much for the valuable suggestion. Yes, the uniform distribution of the mixture powders is important to the quality of the products prepared by LAM. To conformity of the mixtures, we conducted the SEM observations on the mixture powders, and found that the Irregular Nb particles were uniformly distributed around the spherical Ti particles without the agglomeration. The SEM image of the mixtures was not shown in the manuscript for lack of space. In fact, in the previous literature [16] and [17] (see below), the Ti-Nb mixed powders were also obtained by ball-milling method, which are considered to be relatively uniform.

 [16]  Fischer, M., Joguet, D., Robin, G., Peltier, L., Laheurte, P. In situ elaboration of a binary Ti-26Nb alloy by selective laser melting of elemental titanium and niobium mixed powders. Mater Sci Eng C Mater Biol Appl. 2016, 62, 852-859.

[17]  Wang, Q., Han, C., Choma, T., Wei, Q., Yan, C., Song, B., et al. Effect of nb content on microstructure, property and in vitro apatite-forming capability of Ti-Nb alloys fabricated via selective laser melting. Mater. Design. 2017, 126, 268-277.

Point 2. Process parameters are given. Knowing that this is a new powder composition, evidence that these processing parameters are reasonable, a special study needs to be presented.

Response: That is right. To obtain the reasonable processing parameters, we conducted a special study on the effects of the process parameters on the microstructure and properties of the alloy prepared by LAM. The results will published in elsewhere. The parameters adopted in the present work is the optimized ones.

Point 3. Stress-strain curves should be included.

Response: As required, we have added the stress-strain of curves the samples in Figure 12 in the revised manuscript.

Reviewer 2 Report

The authors present vary valuable data on the topic of material mixing in powder additive manufacture. The paper is easy to read and understand. Some comments:

·         Figure 4 – the curves are off-set, which means that the intensity does not increase with annealing temperature, hence the axis title should be relative intensity

·         Figure 5a – a single pore is indicated by the arrow not pores

·         Line 150 – should be attributed not attributable

·         Line 172 – more uniform as compared to what?

·         Figure 9 – I would not agree with the discussion, more grain orientations are shown in figure 9 a than b, which indicates stronger texture of annealed sample  

·         Figure 10- the authors should note that texture depends strongly on the point at which the measurement was taken, more details should be given

·         Figure 12 does not match the texture data, in figure 10 and 11 you show that there is more texture in the annealed specimen at 850 degC but in figure 12 this sample exhibits the best properties (elongation and strength). Could you explain this better?

·         There is not much discussion in the paper, mainly presentation of results  

Author Response

Piont 1. Figure 4 – the curves are off-set, which means that the intensity does not increase with annealing temperature, hence the axis title should be relative intensity.

Response: Thank you very much for your careful review. We have replaced intensity unit with a relative intensity in Figure 4 in the revised manuscript.

Figure 4. XRD patterns of LAM-processed Ti–26Nb alloys after annealing treatment at different temperatures.

Piont 2. Figure 5a – a single pore is indicated by the arrow not pores

Response: Sorry for our carelessness. We have modified Figure 5a (change “pores” into “pore”) in the revised manuscript.

Piont 3. Line 150 – should be attributed not attributable

Response: As required, we have replaced attributable with an attributed in the revised manuscript.

Piont 4. Line 172 – more uniform as compared to what?

Response: That is right, the sentence was modified as: the sample annealed at 650 °C for 0.5 h, elliptical grains began to grow and their size distribution became more uniform (as shown in Fig. 7(a)) as compared to as-deposited sample, which is mark in blue.

Piont 5. Figure 9 – I would not agree with the discussion, more grain orientations are shown in figure 9 a than b, which indicates stronger texture of annealed sample.

Response: Yes, as the reviewer said, the texture depends strongly on the point at which the measurement was taken. The texture strength of the alloy is determined by a statistical results of the measurements in different regions of the samples. The statistical results in the present work indicated that the texture strength of the annealed samples is slightly weaker than that of the as-deposited alloy.

Piont 6. Figure 10- the authors should note that texture depends strongly on the point at which the measurement was taken, more details should be given.

Response: That is right. The Fig.10 is a typical image of the orientation image of the sample. In fact, we conducted the EBSD analysis at the several regions in the sample. The statistical results was give in the experimental report, which was not shown in the manuscript. More details have been given in experimental sections in the revised manuscript.

Piont 7. Figure 12 does not match the texture data, in figure 10 and 11 you show that there is more texture in the annealed specimen at 850 degC but in figure 12. this sample exhibits the best properties (elongation and strength). Could you explain this better? 

Response: In the present work, we found that the annealing treatment has no important on the texture strength of the as-deposited alloy. We think that the microstructure uniformity and grain size has an important effects on the mechanical properties of the alloy. The sample annealed at 850 exhibits more uniform microstructure and relative small grain size, and thus it shows a good comprehensive mechanical properties. It should be noted that the effects of the annealing treatment on the evolution of the texture of the as-deposited alloy and its effects on the mechanical properties of the alloy are very complex. Further work needs to be done in the future.   
